# Psychometric Properties of the Body–Mind–Spirit Wellness Behavior and Characteristic Inventory for the Greek Population

**DOI:** 10.3390/healthcare12040478

**Published:** 2024-02-16

**Authors:** Evangelos Mantsos, Georgιos Lyrakos, Dimitra V. Katsarou, Aglaia Zafeiroudi, Maria Giannousi, Vasiliki Zisi

**Affiliations:** 1Department of Physical Education and Sport Science, University of Thessaly, 42100 Trikala, Greece; v.mantsos@hotmail.com (E.M.); azafeiroudi@uth.gr (A.Z.); 2Psychiatric Ward, General Hospital of Nikaia, 18454 Athens, Greece; geolyr@hotmail.com; 3Department of Preschool Education Sciences and Educational Design, University of the Aegean, 81100 Rhodes, Greece; 4Department of Physical Education and Sport Science, Democritus University of Thrace, 69100 Komotini, Greece; mgiannou@phyed.duth.gr

**Keywords:** whole-person wellness, psychometric testing, aging, language adaptation, quality of life

## Abstract

The Body–Mind–Spirit Wellness Behavior and Characteristic Inventory (BMS-WBCI) is a free-of-charge wellness tool with good psychometric properties, widely used mainly in studies assessing quality of life and healthy lifestyle habits. This certain tool is based on the Hettler’s (1980) model and has been validated for use with students aged 18–36. The purpose of this study was to adapt the BMS-WBCI in the Greek language and at the same time to validate it for use in the general population. This study included 520 participants aged 16–75 (M = 39.86, SD = 10.5), who were recruited from the Greek population using the snowball procedure. The BMS-WBCI was adapted into Greek language, following a multiple forward-and-backward translation protocol. Confirmatory Factor Analysis (CFA) was used to validate the overall construct of the Greek BMS-WBCI. The final solution was a three-factor model with 38 items, after removing the items B1, B8, B9, M11, M24, and S43. This final model demonstrated an acceptable to good fit, presenting higher goodness-of-fit indices (CFI = 0.91, TLI = 0.90) and lower badness-of-fit indices (χ^2^/653 = 2.29, *p* < 0.001, RMSEA = 0.05, SRMR = 0.06). All items in the hypothesized model exhibited statistically significant standardized factor loadings (*p* < 0.001), with loadings consistently above 0.40. A very good internal consistency was found using the composite reliability measures (Body 0.86, Mind 0.95, Spirit 0.94). Further analysis indicated a good convergent validity (average variance extracted values: Body 50.5%, Mind 50.7%, Spirit 54.9%). The results indicated adequate discriminant validity, as all square roots of average variance extracted were higher than the correlation between construct items. In conclusion, this psychometric evaluation of the BMS-WBCI adds to the evidence supporting its use in the Greek language, not only in students, but also in the general population.

## 1. Introduction

The concept of wellness is nowadays an interdisciplinary topic, incorporated in scientific fields such as physical education and sports, tourism, environment, psychology, medicine, and economy [1]. One could think wellness as a way of life that prioritizes good health, balance, and reducing bad habits [2], but in no way should wellness be considered as a synonym of wellbeing. Wellbeing, actually, is a step toward wellness, but it is not as encompassing as wellness [3]. A holistic, self-driven process of wellness is characterized by physical, mental, spiritual, and environmental aspects [4]. There are several theoretical models developed to define the various elements or interrelated areas that comprise wellness. A comparison of these models [5] indicated that there is a total of eight dimensions defining wellness: five of them are common in all models (social, emotional, physical, intellectual, spiritual), and three of them are used interchangeably (psychological, occupational, environmental).

One of these conceptual models for describing wellness is Hettler’s Six Dimensions of Wellness [6]. The six main facets of wellness covered by this hexagonal model include social, emotional, physical, intellectual, spiritual, and occupational components. Hettler’s model’s fundamental premise is that wellbeing is an active, self-driven process. This perspective suggests that an individual becomes conscious of health-related behaviors that either promote or impede their mental, physical, spiritual, and emotional well-being. Once the person is conscious of their actions, (s)he can make decisions that will help attain full potential. Adams’s model conceptualized wellness similarly to Hettler’s, but replaced the occupational dimension with psychological wellness [6,7]. Another popular conceptual framework that explains the traits of and theorizes connections between aspects of healthy functioning is called the Wheel of Wellness (WoW) [8]. The initial form of this model was described as having twelve spokes, but after its evolution, the model focuses on five factors: cognitive, emotional, relational, physical, and spiritual wellness.

The assessment and measurement of wellness was the topic of a quite recent systematic review [3]. In this study, eleven instruments were recognized as being relevant to the topic of wellness, and five of them demonstrated strong reliability values: (a) the Perceived Wellness Survey—PWS, (b) the Wellness Evaluation of Lifestyle—WEL, (c) the Five-Factor Wellness Evaluation of Lifestyle—5F-Wel, (d) the Body–Mind–Spirit Wellness Behavior and Characteristic Inventory—BMS-WBCI, and (e) the Optimal Living Profile—OLP [7,8,9,10,11].

The Perceived Wellness Survey (PWS) is based on the Adams theoretical model [7]. The subscale values’ magnitude (mean) and balance (standard deviation) are added to calculate the composite wellness score [12]. To assess the construct validity of the PWS, a total of 1077 volunteers from three different volunteer groups—college students (n = 281), medical workers (n = 238), and business employees (n = 558)—participated. Adams found that in all but three of the analyses, there was a significant difference between the means of the highest and lowest PWS groups, indicating a high degree of construct validity [12]. The tool has free online access and is easy to use, since it is formed from 36 questions and the administration time is less than 30 min. The composite PWS wellness score, however, is based on perceptions rather than behavioral characteristics, which means that the outcome is a score of wellness perception rather than wellness itself.

The WEL was developed from the theoretical framework of the WoW model and it actually includes all of the dimensions of this model [8]. It has been reported as the gold standard of wellness assessments [13]. The initial version of this instrument however was too long, with 123 items and over 2 h completion time. The 5F-Wel is a newer version of the WEL, with 91 items and substantially improved psychometric properties [9]. There is also reported a newer version of the 5F-well: the 4F-Wel, with only 56 items, but this however is not publicly available [3]. Bart reports that the 5F-Wel and the 4F-Wel are not free to use, since their developers charge a fee.

The BMS-WBCI tool was developed as a wellness evaluation instrument for college students, based on Hettler’s theory [6,10]. As Hey stated, the effort to develop the BMS-WBCI tool was based on the TestWell inventory, which is a test with good psychometric properties; however, it takes too long to administer—over 2 h—and also, it charges a fee for use [10]. The BMS-WBCI takes less than one hour to complete since it has only 44 items, and it was proved to have good psychometric properties (factor split-half reliabilities 0.73 to 0.84, alpha coefficients 0.75 to 0.92). The psychometric properties of this certain instrument were tested in other studies and proved to be of adequate internal consistency for use in college students aged 18–39 [14]. The questions of this instrument, however, are applicable to the general population. After all, the tool from which the BMS-WBCI came from (TestWell inventory) was constructed and tested not only in students but in the general population as well. 

The purpose of the present study was to adapt the BMS-WBCI into the Greek language and at the same time to validate it for use in the general population. It is important to have this instrument validated for the Greek language and culture, since it measures whole-person wellness and is cost-free. It is important to have this instrument validated for the general population, since the research on wellness assessments, at least in the Greek language, is at a very early stage.

## 2. Materials and Methods

### 2.1. Participants

The calculation of the sample size was based on the Greek population aged >15 yrs, which, according to the recent population census, is 8.5 million (Eurostat, https://en.wikipedia.org/wiki/Demographics_of_Greece accessed on 21 December 2023). It was estimated that the required sample size is 850 (1‰), and assuming a 15% missing data allowance, the target sample size was set to 1000. A non-probability convenience sample was recruited from the Greek population, using the snowball recruitment procedure. More specifically, 5 of the co-authors shared the link to the online questionnaire to 50 of their contacts asking them (a) to complete it, (b) share the link to 3 other persons aged 16–64, and (c) share the link to a person over 65 yrs. The online survey was initiated via social platforms such as Facebook and via e-mails, and data were collected using Google forms. The inclusion criteria was as follows: aged over 16, fluently speaks the Greek language, with no comorbid diagnosed psychiatric disorder. In some cases, concerning people of old age with no social media or internet skills, printed questionnaires were also used. 

We collected 643 questionnaires, but 123 of these were incomplete. Thus, the final sample size was 520 people from the Greek general population, comprising 28.8% males and 71.2% females. The mean age of the participants was 39.86 years (SD = 10.5), ranging from 16 to 75 years. Certain demographic characteristics of the participants were recorded, that is, gender, age and marital status, number of children, employment status, education level, monthly income, and place of residence. All of the sample characteristics are depicted in Table 1.

### 2.2. Ethical Considerations

Protection of the human subjects was ensured throughout the study. Prior to data collection, approval to conduct the study was obtained from the ethics committee of the Faculty of Physical Education and Sports Science, University of Thessaly.

Informed consent, following information sheets, was obtained prior to participation in the study. Online consent forms were signed through an accept button, at the end of an information sheet. Specifically, a participant had to accept the terms of the study that complied with the Helsinki declaration, before the administration of the questionnaire. 

Anonymity was maintained during the study as no identifiers were collected on any of the research instruments. Participants were free to withdraw from the study at any moment if they desired, up to the completion of the data collection and start of data analysis. As the data were anonymous, participants needed to use a personal code to request the withdrawal of their data (including 2 letters of their surname and the 3 last digits of their phone number, e.g., YK234).

### 2.3. The Instrument

There are three subscales out of the 44 items in the English version of the BMS-WBCI [10]. The first subscale, dubbed “Body”, comprises 9 items (items 1 through 9) that pertain to risk behaviors in the physical realm of wellness, such as personal safety, physical fitness, and food intake. The intellectual, social, emotional, and occupational dimensions of wellness are represented by 20 items (items 10–29) that make up the second subscale, “Mind”. The final subscale is called “Spirit” and has 15 items (items 30–44) that address emotional, occupational, and spiritual aspects of wellbeing. The participant is asked to rate each of the 44 items on a 3-point Likert scale, with 1 denoting “rare/seldom”, 2 being “occasionally/sometimes”, and 3 being “often/always”. The score is calculated by adding the questions’ ratings, with a total highest possible score of 132 (=3 × 44) and a total lowest score of 44 (=1 × 44). Higher scores indicate a higher degree of engagement in healthy behaviors and agreement with traits that promote overall wellbeing. More specifically, a score in the range of 44–73 indicates: “Needs immediate behavior change to improve wellness lifestyle”, scores ranging from 74 to 103 indicate: “On the way to a wellness lifestyle, but behavior change is needed in certain areas”, and a score in the range of 104 to 132 suggests: “Frequency of behaviors indicates that a healthy lifestyle exists”.

The adaptation of the questionnaire into the Greek language was performed using the multiple forward-and-backward translation protocol [15] following certain criteria [16]. However, due to the diversity of the topics of the questionnaire, the number of translators in the forward translation was increased to five: two physical activity professionals, two health professionals, and one linguistics professional, each with Greek being their mother tongue and an advanced level of English, translated the questionnaire into Greek. During the first reconciliation meeting, all translations were compared, and since there were minimal differences, regarding mainly syntax, a consensus version was obtained. Then, one native English speaker (living and working in Greece for the last 10 years), who was blinded to the original version, retranslated the consensus Greek translation into the source language (back translation), which is the recommended procedure, according to MAPI criteria, for creating semantic equivalence. This is a process of validity checking to make sure that the translated version reflected the same item content as the original version. The two translations (forward and back translations) were compared and differences were found only in item number 1 (“I limit risky behaviors (i.e., driving fast, bungee jumping, parachuting, etc.”). The back translation used the word skydiving instead of parachuting. It was decided to keep the initial translation of parachuting, in order to stick to the original version, although none of these activities are very well known in the Greek reality. 

Pre-testing the translated online version of the instrument on a small group of research team members was the final step in the translation process. This procedure entails asking them in-depth questions concerning how well they understand the questionnaire, and the usability and feasibility of the completion process, and has been proved useful in another study regarding the adaptation of an instrument into the Greek language [17]. 

### 2.4. Data Analysis

A software called SPSS version 26 was used to examine the data. The dataset underwent scrutiny for missing data, normality, linearity, and outliers. The Kolmogorov–Smirnov (K-S) statistic was also taken into account; normality was verified by a non-significant value (*p* > 0.05). Skewness and kurtosis values were produced; values close to zero showed the distribution’s normality. Descriptive statistics were used to evaluate the data obtained from the demographic section of the questionnaire to describe the sample. For the wellness scores, the mean, standard deviation, and range were provided. 

Following the descriptive statistics, the overall construct of the Greek BMS-WBCI was validated through Confirmatory Factor Analysis (CFA) using AMOS 26. Model fit was determined by employing modification indices and covariance procedures. Various goodness-of-fit indices and recommended thresholds [18] were utilized to assess the overall fit of the data model, including Chi-Square (χ^2^), the Chi-Square/degrees of freedom ratio (χ^2^/df ratio), the Comparative Fit Index (CFI > 0.90 considered acceptable and >0.95 considered desirable), the Tucker–Lewis Index (TLI > 0.90 acceptable and >0.95 desirable), Root-Mean-Square Error of Approximation (RMSEA < 0.08 considered an acceptable fit and <0.06 considered a good fit), and Standardized Root-Mean-Square Residual (SRMR <0.08 considered an acceptable fit and <0.05 considered a good fit). Additionally, Akaike Information Criterion, Bayesian Information Criterion, and Consistent Akaike Information Criterion values were considered to compare the fit of alternative models, with lower values indicating a better fit for the hypothesized model.

For assessing internal consistency, composite reliability values greater than 0.70 were deemed indicative of good reliability [19]. Convergent validity of the BMS-WBCI model factors was evaluated through average variance extracted values. To establish convergent validity, average variance extracted values should be equal to or greater than 0.50 and lower than the composite reliability. Discriminant validity was analyzed using maximum shared squared variance, which is accepted when its value is lower than the average variance extracted for each factor. Correlations between factors should be less than 0.80 to establish discriminant validity and equal to or greater than 0.50 to demonstrate convergent validity [19].

## 3. Results

Descriptive statistics regarding the frequencies of given responses of all items of the Greek BMS-WBCI are presented in Table 2. It is apparent that the answers for the items of the Body and Mind items that were ultimately excluded from the final version of the questionnaire are not distributed across all of the available responses, but a large proportion of the answers, in most cases more than half of them, are concentrated in a certain option. On the contrary, in the Spirit scale, the answers are usually concentrated in option 2 or 3, but the answers in the item that was finally excluded (S43) are almost equally distributed across all answer options. 

### 3.1. Confirmatory Factor Analysis

The results of the CFA that was performed to determine the factorial structure of the Greek BMS-WBCI are summarized in Table 3.

The initial three-factor model yielded an inadequate solution (χ^2^/899 = 3.28, *p* < 0.001, CFI = 0.79, TLI = 0.78, RMSEA = 0.07, SRMR = 0.07). To enhance the model’s fit, items “B1”, “B8”, “B9”, “M11”, “M24”, and “S43” were removed based on modification indices from the AMOS program and careful inquiry. Despite these adjustments, the second round of CFA still indicated unsatisfactory goodness of fit (χ^2^/662 = 3.65, *p* < 0.001, CFI = 0.81, TLI = 0.80, RMSEA = 0.07, SRMR = 0.07). However, further refinement by adding error covariances between specific items significantly improved the model’s fit.

In the refined correlated three-factor model, error covariances were introduced between items S33 and S34, B5 and B6, S30 and S31, S36 and S37, M10 and M12, M16 and M20, S35 and S41, M22 and M23, as well as between items S41 and S42. This final model demonstrated an acceptable to good fit, presenting higher goodness-of-fit indices (CFI = 0.91, TLI = 0.90) and lower badness-of-fit indices (χ^2^/653 = 2.29, *p* < 0.001, RMSEA = 0.05, SRMR = 0.06).

The statistical superiority of this refined correlated three-factor model was also confirmed by its considerably lower Akaike Information Criterion, Bayesian Information Criterion, and Consistent Akaike Information Criterion values (AIC = 166.79, BIC = 205.92, and CAIC = 215.12). Regarding item loadings (see Table 4), all items in the hypothesized model exhibited statistically significant standardized factor loadings (*p* < 0.001), with loadings consistently above 0.40. The Spirit factor items demonstrated relatively higher reliability estimates (0.55, 0.55, 0.48, 0.57, 0.55, 0.58, 0.52, 0.61, 0.54, 0.61, 0.47, 0.61, 0.52, 0.51) compared to items in other factors, as evidenced by the squared value of each standardized factor loading. These findings strongly support the adequacy of this Greek BMS-WBCI three-factor model.

### 3.2. Validity and Reliability

Following the model’s fit, the validity and reliability of the Greek BMS-WBCI three-factor model was evaluated. Composite reliability served as a measure of internal consistency for latent variables, where values above 0.70 indicated good reliability [19]. The first latent variable, representing the Body (items B2, B3, B4, B5, B6, and B7), showed a composite reliability of 0.86. The second latent variable, representing the Mind (items M10–M29), had a composite reliability of 0.95. The third latent variable, representing the Spirit (items S30–S44), demonstrated a composite reliability of 0.94.

Convergent validity was assessed using average variance extracted values, with values equal to or greater than 0.50 (and less than the composite reliability) indicating convergent validity [19]. The average variance extracted values were 50.5% for Body, 50.7% for Mind, and 54.9% for Spirit, all exceeding the 50-percent benchmark and falling below the respective composite reliability.

Discriminant validity was examined through maximum shared squared variance, which is acceptable when its value is lower than the average variance extracted value for each factor [19]. The Greek BMS-WBCI three-factor model demonstrated validity in terms of discriminant validity, as the maximum shared squared variance values for individual constructs were lower than their respective average variance extracted estimates.

Additionally, discriminant validity was assessed by comparing the square root of the average variance extracted values with the correlation between the construct items. Adequate discriminant validity requires the square root of the average variance extracted to be higher than the correlation between latent variables [19,20,21,22]. For the Body latent variable, the square root of the average variance extracted was 0.71, surpassing the correlation values for Mind (0.262) and Spirit (0.328). Similarly, for the Mind latent variable, the square root of the average variance extracted was 0.71, exceeding the correlation values for Body (0.262) and Spirit (0.645). Finally, for the Spirit latent variable, the square root of the average variance extracted was 0.74, surpassing the correlation values for Body (0.328) and Mind (0.645). In summary, the results indicated adequate discriminant validity, as all square roots of the average variance extracted values were higher than the correlation between construct items.

## 4. Discussion

In the present study, the 44-item Body–Mind–Spirit Wellness Behavior and Characteristic Inventory (BMS-WBCI) was adapted into the Greek language, following a certain translation protocol and criteria for cultural adaptation [15,16]. It was validated for use in the general population, whereas studies on the initial version of the questionnaire provided psychometric properties only for students aged up to 36 years [10,14]. The third round of CFA produced a 38-item Greek BMS-WBCI three-factor model, with good psychometric properties. The above model includes 6 items for the Body dimension, 18 items for the Mind dimension, and 14 for the Spirit dimension. The 38-item Greek BMS-WBCI exhibited excellent internal consistency (Body 0.86, Mind 0.95, Spirit 0.94), good convergent validity, and adequate discriminant validity. These results are consistent with Hey’s initial findings [10] and imply that these three components were sufficiently stable for use as a subscale for each of the three dimensions.

The Greek BMS-WBCI is shorter than the original version by six items, which were excluded as they seemed to not go well with other wellness measures. These items may be attributed to the health behaviors and the culture of the Greek population. More specifically, the Body dimension in the Greek version of the questionnaire is shorter by three items: B1—I limit risky behaviors (i.e., driving fast, bungee jumping, parachuting, etc.), B8—I drink at least eight glasses of water a day, and B9—I surround myself with physically healthy people. The item B1 seemed to be problematic even in the translation procedure, since activities like parachuting, skydiving, and bungee jumping are not common in Greece as leisure activities. The majority of the participants (65%) noted that they often or always limit risky behaviors. This danger avoidance was probably not linked to the age of the participants, since 68% of them were aged below 45, which is quite young to suggest that type of socio-emotional effect on health behaviors. The above findings are probably linked to the socio-cultural habits of the Greek population, which does not seem to enjoy risky behaviors. The answers to item B8 were also concentrated within the option “often/always” (48.7%), and this result should also be attributed to socio-cultural habits. There is plenty of water in Greece, and tap water is almost always of good quality. A glass of water is always a companion of a treat at home or a patisserie. At cafés, a glass of water arrives at your table before you place your order, and take-away coffee, most of the times, is accompanied by a free-of-charge small bottle of water. Drinking water might not be a health behavior of which these individuals need to become conscious [6], but rather it is a habit of Greek people. Finally, physical health is probably not a criterion of social and personal relations, since the majority of the participants (57.5%) for the question B9 picked the option occasionally or sometimes.

The Mind dimension of the Greek BMS-WBCI is shorter by two items: M11—I am open to new ideas, and M24—I am tolerant of others whether or not I approve of their behavior or beliefs. Both questions are linked to openness and the majority of the participants picked the answers often or always (M11—794%, M24—57.5%). Although being open minded is an intellectual component of wellness, it seems that these people do not have to become aware of how to train themselves in openness, since their culture and their society already train them to be open-minded to new ideas and accept different people with different attitudes and habits. Finally, the Spirit dimension missed one item in the Greek version: S43—I read some form of spiritual literature on a regular basis. One look at the frequencies of the Spirit dimension makes it clear that the only item with a frequency of “rare/seldom” responses around 33% is the S43 variable. It is clear that 1/3 of the participants are not used to reading spiritual literature; however, the percentage of people that picked the response “rare/seldom” in all of the other items of the Spirit variable is very small. In most items, around 90% of the responses are spread among the options “occasionally/sometimes” and “often/always”. The positive attitudes of the participants in the Spirit items might be linked to their religion: 90% of Greeks are Greek Orthodox (https://en.wikipedia.org/wiki/Religion_in_Greece, accessed on 21 December 2023), and although spirituality is not religiousness, these two concepts might be close in this certain religion.

Finally, some limitations should be taken into consideration, including the small sample size and the age group proportions. We intended to have 850 participants in our study but the final sample size was 520. The age groups in our sample do not correspond to the age structure reported by official demographics (https://en.wikipedia.org/wiki/Demographics_of_Greece#Age_structure, accessed on 21 December 2023). The percentages of the population in the age groups 25–54 and 55–64, according to recent demographics, are 42.45% and 13.13%, respectively. In our study, the percentages of the participants in these age groups were 60.8% and 30.8%, respectively. On the other hand, the age group over 64 is underrepresented in our study, since the official demographics report that people of this age make up 20.91% of the population, and in our study, they were just 1%. The above percentages probably limit our findings for the ages below 65.

There are also some limitations regarding other characteristics of the participants. Almost all of the participants in this study identified as Caucasian, and half of the sample lived in urban areas, limiting the generalizability of the results to the entire country. In terms of demographics, the sample was slightly over-representative of females. In previous studies that used the BMS-WBCI, the study populations identified as dominantly Caucasian, with nearly equal numbers of female and male participants and a slightly under-representation of the percentage of females [10,15].

## 5. Conclusions

In conclusion, this study adds to the evidence that the BMS-WBCI can be used to assess wellness not only in students, but also in the general population. Further research is needed with a larger sample size and a greater proportion of older adults. The BMS-WBCI is a free-to-use instrument with good psychometric properties, measuring the main dimensions of wellness. Since wellness is an interdisciplinary topic, there are many research areas and scientific fields that could benefit from this tool: for example, people who work with therapeutical models, those who work in medicine, social workers, etc. It should also be interesting to test the psychometric properties of the BMS-WBCI in different cultures. The wellness differences between different cultures could provide useful information on education and training practices for improving healthy habits and wellness across diverse cultures all over the world.

## Figures and Tables

**Table 1 healthcare-12-00478-t001:** Sociodemographic characteristics of the participants.

		Frequency	Percent %	Cumulative Percent %
Gender	Males	150	28.8	28.8
	Females	370	71.2	100
Educational Level	Elementary School	5	1	1
	High School	53	10.2	11.2
	Bachelor	199	38.3	49.4
	M.Sc.–Ph.D.	263	50.6	100
Employment Status	Public Employment	209	40.2	40.2
	Private Employment	137	26.3	66.5
	Private Practice	72	13.8	80.4
	Unemployed	63	12.1	92.5
	Student	28	5.4	97.9
	Pensioner	11	2.1	100
Place of Residence	Large Urban Center	303	58.3	58.3
	District Town	162	31.2	89.4
	Village/Small Town	55	10.6	100
Age group	16–17	5	1.0	1
(yrs)	18–24	34	6.5	6.6
	25–54	316	60.8	68.0
	55–64	160	30.8	99.0
	65+	5	1.0	100.0
	Total	520	100.0	

**Table 2 healthcare-12-00478-t002:** Answer frequencies for all items of the Greek BMS-WBCI (# items excluded from the final version).

	Frequency (%)		
	1—Rare/Seldom	2—Occasionally/Sometimes	3—Often/Always
Body			
# B1. I limit risky behaviors (i.e., driving fast, bungee jumping, parachuting, etc.).	62 (11.9)	120 (23.1)	338 (65.0)
B2. I maintain my fitness by exercising regularly and maintaining my weight.	94 (18.1)	244 (46.9)	182 (35.0)
B3. I have a reasonable amount of flexibility and do exercises that help maintain my range of motion.	121 (23.3)	247 (47.5)	152 (29.2)
B4. I use warm-up activities before exercising to help prevent injuries.	128 (24.6)	173 (33.3)	219 (42.1)
B5. I eat a variety of foods and consume the recommended number of servings from each food group.	68 (13.1)	213 (41.0)	239 (46.0)
B6. I eat a balanced diet low in saturated fat and cholesterol.	84 (16.2)	261 (50.2)	175 (33.7)
B7. I participate in recreational sports or activities that help maintain my fitness.	180 (34.6)	215 (41.3)	125 (24.0)
# B8. I drink at least eight glasses of water a day.	98 (18.8)	169 (32.5)	253 (48.7)
# B9. I surround myself with physically healthy people.	32 (6.2)	299 (57.5)	189 (36.3)
Mind			
M10. I learn from my past life experiences.	10 (1.9)	153 (29.4)	357 (68.7)
# M11. I am open to new ideas.	9 (1.7)	98 (18.8)	413 (79.4)
M12. I learn from my mistakes and try to behave differently the next time.	11 (2.1)	122 (23.5)	387 (74.4)
M13. I talk with people rather than talk at people.	16 (3.1)	119 (22.9)	385 (74.0)
M14. I accept responsibility for my actions.	11 (2.1)	83 (16.0)	426 (81.9)
M15. I understand and accept the existence of cultural diversity and its contribution to the quality of life.	12 (2.3)	91 (17.5)	417 (80.2)
M16. I make good ethical decisions.	13 (2.5)	157 (30.2)	350 (67.3)
M17. I consider alternatives before making decisions.	13 (2.5)	118 (22.7)	389 (74.8)
M18. I focus on reality.	14 (2.7)	126 (24.2)	380 (73.1)
M19. I am flexible to changes and can maintain stability in my life in healthy ways.	19 (3.7)	235 (45.2)	266 (51.2)
M20. I have strong morals and healthy values.	10 (1.9)	117 (22.5)	393 (75.6)
M21. I learn from the mistakes of others.	21 (4.0)	218 (419)	281 (54.0)
M22. I have satisfying interpersonal relationships.	16 (3.1)	181 (34.8)	323 (62.1)
M23. I feel loved and supported by family and friends.	17 (3.3)	116 (22.3)	387 (74.4)
# M24. I am tolerant of others whether or not I approve of their behavior or beliefs.	17 (3.3)	204 (39.2)	299 (57.5)
M25. I set achievable goals for myself.	21 (4.0)	205 (39.4)	294 (56.5)
M26. I handle various social settings well.	9 (1.7)	212 (40.8)	299 (57.5)
M27. I analyze my thoughts (I think, question, and evaluate) before I act.	14 (2.7)	144 (27.7)	362 (69.6)
M28. I make the best of bad situations.	17 (3.3)	226 (43.5)	277 (53.3)
M29. I express my feelings with others and consider their feelings.	34 (6.5)	178 (34.2)	308 (59.2)
Spirit			
S30. I experience harmony within.	52 (10.0)	290 (55.8)	178 (34.2)
S31. I experience peace of mind.	54 (10.4)	278 (53.5)	188 (36.2)
S32. I am in touch with the soul within.	45 (8.7)	215 (41.3)	260 (50)
S33. I experience happiness within.	38 (7.3)	276 (53.1)	206 (39.6)
S34. I experience joy within.	30 (5.8)	265 (51.0)	225 (43.3)
S35. I experience self-satisfaction.	31 (6.0)	260 (50.0)	229 (44.0)
S36. I express my spirituality appropriately and in healthy ways.	32 (6.2)	230 (44.2)	258 (49.6)
S37. My spirituality helps me remain calm and strong and helps me to better deal with difficult times.	32 (6.2)	206 (39.6)	282 (54.2)
S38. I recognize the positive contribution faith can make to the quality of my life.	61 (11.7)	190 (36.5)	269 (51.7)
S39. I routinely undertake new experiences to enhance my spiritual health.	54 (10.4)	234 (45.0)	232 (44.6)
S40. I have a positive outlook on life.	19 (3.7)	197 (37.9)	304 (58.5)
S41. I am content with who I am.	30 (5.8)	223 (42.9)	267 (51.3)
S42. I know my purpose in life.	62 (11.9)	217 (41.7)	241 (46.3)
S43. I read some form of spiritual literature on a regular basis.	163 (31.3)	197 (37.9)	160 (30.8)
S44. I experience love of others and myself.	24 (4.6)	191 (36.7)	305 (58.7)

**Table 3 healthcare-12-00478-t003:** Fit indices for the three-factor model of the BMS-WBCI.

Models	*χ^2^*	*df*	*χ^2^*/df	CFI	TLI	RMSEA	SRMR	AIC	BIC	CAIC
1st round	2950.58 *	899	3.28	0.79	0.78	0.07	0.07	313.26	351.97	361.07
2nd round	2417.40 *	662	3.65	0.81	0.80	0.07	0.07	257.54	291.15	299.05
3rd round	1483.85 *	653	2.29	0.91	0.90	0.05	0.06	166.79	205.92	215.12

Note: N = 103; *χ^2^* = Chi-Square goodness-of-fit statistic; *df* = degrees of freedom; *χ^2^*/*df* = Chi-Square per degree of freedom; CFI = Comparative Fit Index; TLI = Tucker–Lewis Index; RMSEA = Root-Mean-Square Error of Approximation; SRMR = Standardized Root-Mean-Square Residual; AIC = Akaike Information Criterion; BIC = Bayesian Information Criterion; CAIC = Consistent AIC. * Indicates that *χ^2^* is statistically significant (*p* < 0.05).

**Table 4 healthcare-12-00478-t004:** Standardized regression weights, squared multiple correlations, and error variance of the Greek BMS-WBCI three-factor model.

Construct/Item	Beta	R^2^	Error Variance	Construct/Item	Beta	R^2^	Error Variance
Body							
B2	0.77 *	0.59	0.41	M25	0.70 *	0.49	0.51
B3	0.68 *	0.46	0.54	M26	0.75 *	0.56	0.44
B4	0.71 *	0.5	0.5	M27	0.67 *	0.45	0.55
B5	0.70 *	0.49	0.51	M28	0.75 *	0.57	0.43
B6	0.72 *	0.51	0.49	M29	0.72 *	0.51	0.49
B7	0.69 *	0.48	0.52	Spirit			
Mind				S30	0.74 *	0.55	0.45
M10	0.72 *	0.52	0.48	S31	0.74 *	0.55	0.45
M12	0.68 *	0.47	0.53	S32	0.69 *	0.48	0.52
M13	0.74 *	0.54	0.46	S33	0.75 *	0.57	0.43
M14	0.73 *	0.53	0.47	S34	0.74 *	0.55	0.45
M15	0.72 *	0.52	0.48	S35	0.76 *	0.58	0.42
M16	0.80 *	0.63	0.37	S36	0.72 *	0.52	0.48
M17	0.71 *	0.51	0.49	S37	0.78 *	0.61	0.39
M18	0.67 *	0.44	0.56	S38	0.74 *	0.54	0.46
M19	0.66 *	0.43	0.57	S39	0.78 *	0.61	0.39
M20	0.72 *	0.51	0.49	S40	0.69 *	0.47	0.53
M21	0.71 *	0.5	0.5	S41	0.78 *	0.61	0.39
M22	0.72 *	0.52	0.48	S42	0.72 *	0.52	0.48
M23	0.63 *	0.4	0.6	S44	0.72 *	0.51	0.49

* Note: Beta *=* standardized regression weights; R^2^ = squared multiple correlations; * *p* < 0.001.

## Data Availability

All data are available upon reasonable request.

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
