# Peer review of "Psychometric Properties of the Body–Mind–Spirit Wellness Behavior and Characteristic Inventory for the Greek Population"

_healthcare, 2024, doi:10.3390/healthcare12040478_

Round 1
Reviewer 1 Report
Comments and Suggestions for Authors
Thank you for the opportunity to review the manuscript titled ' Psychometric properties of the Body-Mind-Spirit Wellness Behavior and Characteristic Inventory for the Greek population ' After a thorough evaluation, I wish to share several critical observations. This study involves translating the BMS-WBCI into Greek and testing its reliability and validity. Although the translation process is rigorous, it appears to lack a more comprehensive evaluation of the scale's psychometric properties. It's particularly notable that confirmatory factor analysis (CFA) is missing, and criterion validity of the scale has not been tested. I understand that the authors may not have collected data on other variables, but considering that CFA is a predominant approach for testing factorial validity – and given that the scale was not developed by the authors, thus its factor structure should be clear – CFA is deemed necessary.
1. Introduction Section - BMS-WBCI Application: I suggest that the introduction should include more details about the application of the Body-Mind-Spirit Wellness Behavior and Characteristic Inventory (BMS-WBCI), particularly incorporating information about the psychometric properties of this scale, beyond the data provided by Hey et al. This would offer a broader understanding of the tool's relevance and application.
2. Sample Size Justification: It's crucial to elaborate on how the sample size was determined and to justify how this number is representative of the entire Greek population. This would strengthen the validity of the study's findings.
3. Sampling Strategy - Snowballing Method: The manuscript should clearly describe the use of the snowballing strategy to reach potential samples. This is particularly important for understanding how the survey was conducted among elderly populations, a group that might have unique challenges in terms of accessibility and response.
4. Heterogeneity of Data Collection: The authors need to explain their statement about collecting a "good amount of heterogeneous data in about 20 days." Clarification is needed on how they ensured the data was heterogeneous and representative.
5. Clarification on Hey et al.'s Scoring Grid: The use of Hey et al.'s scoring grid for grouping scores is not clear. A more detailed explanation is needed to understand how this grid was applied to distinguish between general wellness practices in the sample.
6. Data Analysis Strategy - Factorial Validity: There are concerns about the chosen data analysis methods. Typically, principal axis factoring is more suitable for testing factorial validity than principal component analysis. Additionally, given that the subscales are interrelated, orthogonal rotation might not be the most appropriate choice. The manuscript should address these concerns.
7. Conducting Confirmatory Factor Analysis (CFA): Since the scale wasn't developed by the authors themselves, conducting a confirmatory factor analysis (CFA) could be beneficial. CFA would test the structure of the scale and provide additional evidence of its validity. The authors should consider including this in their analysis.
Overall, these points focus on enhancing the clarity, methodology, and analytical rigor of the study, ensuring that the results are valid, reliable, and applicable to the intended population.
Reviewer 2 Report
Comments and Suggestions for Authors
Please see attached

Reviewer 3 Report
Comments and Suggestions for Authors
I enjoyed reading this manuscript as I found it substantively interesting, and with likely value for practitioners and researchers. I do have several concerns which I express below. Hopefully, my thoughts are thoughtful if not helpful with revision.
NTRODUCTION.
Subheads would add a framework structure. The introduction provides a good summary of some relevant 'wellness' scales and why you have focused your research. I learned very much from reading the Introduction. My suggestion is that you consider using subheads to organize the critical issues.
1. Why is assessing 'wellness' important?
2. Why is the adaptation appropriate? (language)
a. clinical and research value
b. if age, then hypothesize differential age effects on wellness factors
3. What is the purpose of this research?
a. adaptation in another language
b. abstract 'further validate it for use in a population of wider age range.' Why
Note, I consider use in with different age groups as possible 'adaptation' if it requires changes to the instrumentation and not only sampling and analysis.
Presumably, the only adaptation to items and administration are language. And, the 3 scales are comprised of items which are consistent across all respondents irrespective of age and demographic, therefore, hypotheses regarding differences should be reasonable, e.g., age differences with respect to the 'body' scale, which "comprises nine items . . . that pertain to risk behaviors in the physical realm of wellness, such as personal safety, physical fitness, and food intake." A similar argument is possible with respect to the remaining two scales ('mind', 'spirit'). This should be clear. Note, also, if no changes are made to items other than language, then growth models (hypotheses) seem reasonable.
If this is correct, the following suggestions will improve the manuscript and research.
1. provide explicit hypotheses in either the introduction or methods regarding group effects with respect to each scale, particularly for some reasonable age brackets.
2. While extensive and perhaps beyond the scope of what you want to do, you should have completed some differential item functioning analyses to determine if items function differentially across groups, in which case group differences are uninterpretable. NOTE, this is what should have been done with the language adaptation as is a routine analysis when translations are completed. Please consider standards provided by the International Test Commission ((ITC https://www.intestcom.org/)
METHODS.
Sampling
1. How is sample representative of the Greek general population? Offer some statistical tests comparing your statistics to the national parameters, and refer to demographics table (Perhaps the table excessive.)
Adaptation
1.Procedures (language, age)
a. Make it clear that English is the native language and that while the sampling does expand age, no changes to items take age or other demographics into consideration, i.e., translation is consistent across demographics. Perhaps refer to ITC guidelines along with your reference.
Validation
1. The analyses are relatively simplistic as they test hypotheses regarding age differences on the 3 scales. Specify the hypotheses and statistical procedures. Also, other hypotheses should be specified (gender, race, etc.). Perhaps you have interactions to test. The differential item functioning (DIF) analyses should you do them would add to the validation as they would support any group differences you obtain, and without the DIF analyses, the group differences are questionable. This may be true for specific items only, i.e., those items for which age groups may have semantically or substantively different interpretations of the item stems or response options (Likert scales).
2. The correlations matrix of the 3 scales is well described. This should be invariant across groups, i.e., though means and sds may vary, the correlations should remain stable. If they are not invariant, what does that mean? Should you test for invariance, and what tests of invariance are important. You specify so many demographics, most of which are irrelevant for this matter, but some are of course important, e.g., age. What are the age brackets?
Psychometric modeling
1. The descriptions of Cronbach coefficient alpha and split half reliability are good, but insufficient for thoroughly modeling item and scale level psychometrics. Perhaps they will suffice for this manuscript, though I think more can be offered with the data you have.
2. IRT differential item functioning would be appropriate
3. SEM would be preferred to the simplistic principal components estimation with varimax rotation and correlation among factors computations provided by the authors. With the strong structure, a structural equation model would be a stronger hypothesis testing approach, at least more compelling scientifically.
RESULTS.
I suggest introducing the Results with a restatement of purpose and framing the presentation. The need for a Wellness scale, specifically for clinical application and for a broad age range would help with presentation. I do not think Table 2 is necessary unless you provide subgroup means. I suggest providing the reliability estimates (split half and alpha) in narrative for each of the three scales.
Round 2
Reviewer 1 Report
Comments and Suggestions for Authors
I appreciated the author's revision. I recommend publishing the current version.
Reviewer 2 Report
Comments and Suggestions for Authors
accept in present form